# Changes in infant head shape: Developmental trends during the first year of life and secular changes observed in recent years

Eujin Lee[1]*, Hama Watanabe[1], Ryoya Saji[2,3], Fumitaka Homae[4,5], Gentaro Taga[1]

**1** Graduate School of Education, The University of Tokyo, Tokyo, Japan, **2** College of Agriculture, Tamagawa University, Tokyo, Japan, **3** Brain Science Institute, Tamagawa University, Tokyo, Japan, **4** Department of Language Sciences, Tokyo Metropolitan University, Tokyo, Japan, **5** Research Center for Language, Brain and Genetics, Tokyo Metropolitan University, Tokyo, Japan

* elee715@p.u-tokyo.ac.jp

## Abstract

The cephalic index (CI), defined as the percentage of head width to length, has been used in multiple studies. However, CI only provides 2D information, and comprehensive methods to assess the 3D cranial shape are yet to be established. Currently the typical pattern of changes in the head shape in healthy infants remains to be determined. There are only a few studies that followed the changes during the first year of life at frequent intervals or investigated differences between the subpopulations. In this retrospective cross-sectional study, we aimed to identify the morphological changes in the head that occur during the first year of life in healthy Japanese infants and capture differences in relation to background characteristics including age and birth year. We used 1,980 records of measurement data including over the head distances from the left tragion to the right tragion and from the glabella to the occipital protuberance, and head circumference. Complete data across three measurements were available for 909 records. From these measurements, head length, width, and height were estimated to determine CI, volume and a head roundness measure, Globularity Index (GI), by ellipsoid approximation. During the first year, the estimated marginal mean of CI reached the highest level 100.57 (95% CI [97.64, 103.50]) at 6 months (Cohen's $d$: 0.88–1.07, $p < 0.001$), displaying front-to-back flattening before declining to 90.12 (95% CI [85.22, 95.02]) at 12 months following the effect of age, controlling for sex and birth year. We also observed secular changes during the period from 2010 to 2019, with recent birth years presenting more elongated head shape at 3 months of age. These indicated that distinct morphological changes in the head which may result from different growth rates in specific regions occur and secular changes can be observed during the early periods of infancy.

**Data availability statement:** All relevant data are within the paper and its Supporting Information files.

**Funding:** The study was partly supported by Japan Society for Promotion of Science Grants-in-Aid for Scientific Research (23H05425) to G.T. There was no additional external funding received for this study.

**Competing interests:** The authors have declared that no competing interests exist.

## Introduction

The most significant period of brain growth occurs in the first year of life, with the cranial volume nearing about 90% of the adult volume [1]. The cephalic index (CI), which is defined as the ratio between head width and length expressed as a percentage, has been utilized in multiple studies as an objective measure to monitor cranial growth in healthy infants and infants with cranial deformities [2]. However, the CI norms were largely established in the 1980s, and the latest studies suggest that the standards should be updated to reflect the modern populations that seem to drift from the established norm, additionally considering ethnic variability [3–5]. Besides, a few recent studies indicated that not only CI norms vary but also the peak of CI may be attained at different times during the first year of life in healthy infants, depending on the ethnicity [4,6]. For these reasons, it is more desirable to investigate the morphological changes in a specific ethnic group and developmental stage.

Currently, there is no agreement on the typical pattern of morphological development in healthy infants, partly because comprehensive methods to examine the 3D cranial shape are yet to be established [7]. Moreover, as the CI only involves two dimensional (2D) measurements in the horizontal plane, it cannot be used to adequately evaluate the cranial shape which is in three dimensions (3D), which leads to medical practitioners depending on subjective evaluation such as appearance for the assessment of morphological improvements after medical or non-medical interventions [8].

In recent years, several methodologies have been devised to characterize normal cranial development during the initial years of life, engaging various types of equipment including tape measures, calipers, CT and MRI scans and 3D photogrammetric scanners.

Of these, the most widely used tools to assess the head shape in the clinical setting on top of visual inspection are calipers and tape measures, as they are easily applied to infants in a short time frame [9]. Tape measures are used to capture cephalic circumference, and calipers may be directly applied to the infant's head to measure the distance from the glabella to the most protruding point of the occipital region for the head length and the distance between the largest parietal prominences for the head width [10]. Nonetheless, these manual measures provide limited information regarding the 3D profile and are likely to be affected by intra- and inter-rater variations.

To capture 3D and cross-sectional head images, many of the earlier studies relied on MRI and CT scanners [11–15]. Yet, as CT scanning necessitates radiation exposure and both MRI and CT scanning take a long time for recording and require infant sedation, recruiting a large number of healthy infants to create a database or conducting longitudinal monitoring can be challenging. Meanwhile, recent advancements in 3D stereophotogrammetry, which may involve the use of multiple cameras or scanners that are non-invasive, allow for easier and quicker application [16–21]. However, the high cost of the camera devices still prevents them from being broadly used. Today, low-cost smartphone-based solutions that employ smartphone cameras and accessories and are designed specifically to generate 3D models of an infant's head are on their way to wider application [16–18].

Although there are several options available for the evaluation of infant head shapes, the ideal method to be implemented should be easy, inexpensive, non-invasive and feasible in the everyday contexts without the need for additional accessories or steps. Especially, methods involving tape measures are advantageous in capturing changes in the head shape in the long term, and their wide use in the clinical practice indicates abundance of such measurement data. In addition, there are a limited number of studies that focus on the natural course of head growth in healthy infants during the first year of life. Furthermore, only a few studies record the changes at frequent intervals during the first year of life to provide a detailed view or elucidate differences between the subpopulations based on birth year to visualize secular changes that are reported to be occurring in adults and children [19,20].

In this retrospective cross-sectional study, we strived to ascertain the morphological changes in the head in healthy infants that occur in the first year of life, while focusing on the specific ethnic group. For this, we used the existing database comprising 1,980 records of head shape measurements from healthy Japanese infants. These measurements included head circumference (HC), over-the-head distances from the left tragion to the right tragion (LT-RT) and glabella to occipital protuberance (G-O), all of which were obtained using only tape measures. From these direct measurements, the head length, width, and height were estimated to determine the CI and volume by ellipsoid approximation. We sought to identify morphological differences in the typical head shape in relation to background characteristics including age, sex and birth year. In addition, we developed a method which evaluates the 3D roundness of the head to complement the CI by providing additional objective 3D information.

## Materials and methods

### Data source

This study used a database containing records of head shape measurements and background characteristics of infants who participated in various laboratory experiments that took place at the Developmental Brain Science Laboratory, Graduate School of Education, The University of Tokyo during the period from 1st Nov 2006–20th Feb 2020. The recorded head shape measurements included the following three: Head circumference (HC) was measured by encircling the head along a horizontal plane passing through the glabella (G) and the occipital (O) protuberance. Left tragion to right tragion (LT–RT) distance was measured along the surface of the scalp, passing through the left tragion, Cz (as defined according to the international 10–20 EEG system), and the right tragion. Glabella-Occipital (G-O) distance was measured along the surface of the scalp, passing through the glabella protuberance (G), Cz, and occipital protuberance (O). All these direct measurements were collected using tape measures.

Background characteristics included birth weight, birthday, sex, expected delivery date, and date of visit. Differences between birthday, date of visit, and expected delivery date in days were obtained through subtraction. Preterm infants born more than 21 days prior to the expected due date were not included in this study and birth weights were not considered when defining preterm. Only healthy full-term infants with no known neurological disorders at measurement were included in the analysis.

### Data characteristics

The database included a total of 1,980 records, with 1,038 records of males and 942 of females (Table 1 (a)). All records included head circumference (HC) measurements. Each record was considered as a unique infant although there were records corresponding to follow-up visits by the same individuals. Of the records, 1,853 records corresponded to first visit, 95 to second, 8 to third, 6 to fourth, fifth and sixth, 4 to seventh and 2 to eighth visit. Most follow-up visits were second visits; therefore, most multiple records were duplicates. The birth weight ranged from 1,774–4,505 grams, with a mean weight of 3,056 grams and a median weight of 3,044 grams. Compete background information on age, sex, birth year and birth weight were available for all records except two, for which data on birth weight were unavailable. The birth years spanned from 2006 to 2019. The age ranged from 47 to 470 days (or 6.71 to 67.14 weeks).

**Table 1. Summary of background characteristics. (a) all infants, (b) infants with glabella-occipital protuberance (G-O), left tragion-right tragion (LT-RT), and head circumference (HC) measurements, excluding age groups other than 2, 3, 6, and 12 months, (c) the number of infants with G-O, LT-RT, and HC data in each age group and (d) the number of 3-month-old infants with G-O, LT-RT, HC data in each birth year group.**

**(a) All infants**

|  | All | Male | Female |
|---|---|---|---|
| No. of infants (n) | 1,980 | 1,038 | 942 |
|  | Mean | Median | Range |
| Age (days) | 118 | 109 | 47~470- |
| Birth weight (g) | 3,056 | 3,044 | 1,774~4,505 |
| Birth year | – | – | 2006~2019 |

**(b) Infants with G-O, LT-RT, and HC data**

|  | All | Male | Female |
|---|---|---|---|
| No. of infants (n) | 909 | 483 | 426 |
|  | Mean | Median | Range |
| Age (days) | 119 | 106 | 64~389 |
| Birth weight (g) | 3,054 | 3,040 | 1,774~4,505 |
| Birth year | – | – | 2010~2019 |

**(c) The number of infants with G-O, LT-RT, and HC data in each age group.**

|  | 2 months | 3 months | 6 months | 12 months |
|---|---|---|---|---|
| All | 207 | 573 | 100 | 29 |
| Male | 103 | 298 | 63 | 19 |
| Female | 104 | 275 | 37 | 10 |

**(d) The number of 3-month-old infants with G-O, LT-RT, HC data in each birth year group**

|  | 2010 | 2011 | 2012 | 2013 | 2014 | 2015 | 2016 | 2017 | 2018 | 2019 |
|---|---|---|---|---|---|---|---|---|---|---|
| All | 52 | 100 | 148 | 25 | 52 | 15 | 19 | 51 | 63 | 48 |
| Male | 27 | 52 | 81 | 12 | 38 | 8 | 9 | 24 | 31 | 16 |
| Female | 25 | 48 | 67 | 13 | 14 | 7 | 10 | 27 | 32 | 32 |

Out of the 1,980 records, 946 records contained not only head circumference (HC) but also glabella-occipital protuberance (G-O) and left tragion-right tragion (LT-RT) measurements. Most of these records with additional LT-RT and G-O measurements corresponded to infants who were either 2, 3, 6 or 12 months old. Therefore, we analysed 909 records (483 males and 426 females) representing infants aged 2, 3, 6, or 12 months with a complete set of HC, G-O and LT-RT measurements (Table 1 (b)-(c)). Of these, 20 records were the second visit and 1 was the third visit. Most of the follow-up visits did not involve the complete set of measurements in these age groups. All records were grouped into monthly categories based on their age. For instance, one month period was considered as 30 days and records with an age falling between 61–90 days were classified as 2 months old. The number of infants for each age group is also detailed in Table 1 (c).

## Ethical considerations

All participants were full-term infants recruited via the local Basic Resident Register. Ethical approval for this study was obtained from the ethical committee of Life Science Research Ethics and Safety, the University of Tokyo, and written informed consent was obtained from the parent(s) of all the infants prior to the initiation. The authors had access to the individual identifiable information during and after data collection and the database became accessible to the authors for the research purpose on 10th June 2024.

## Cephalic Index and head volume estimation

For estimation of CI, head widths, lengths and heights were first deduced from the three direct measurements (HC, G-O, and LT-RT) as shown in Fig 1 under the assumption that the head resembles an ellipsoid as previously reported [21,22]. When

$$m_0 = HC \tag{1}$$

$$m_1 = LT\text{-}RT \tag{2}$$

$$m_2 = GO \tag{3}$$

$$a_1 = \text{Head width} \times 1/2 \tag{4}$$

$$a_2 = \text{Head length} \times 1/2 \tag{5}$$

$$h = \text{Height} \tag{6}$$

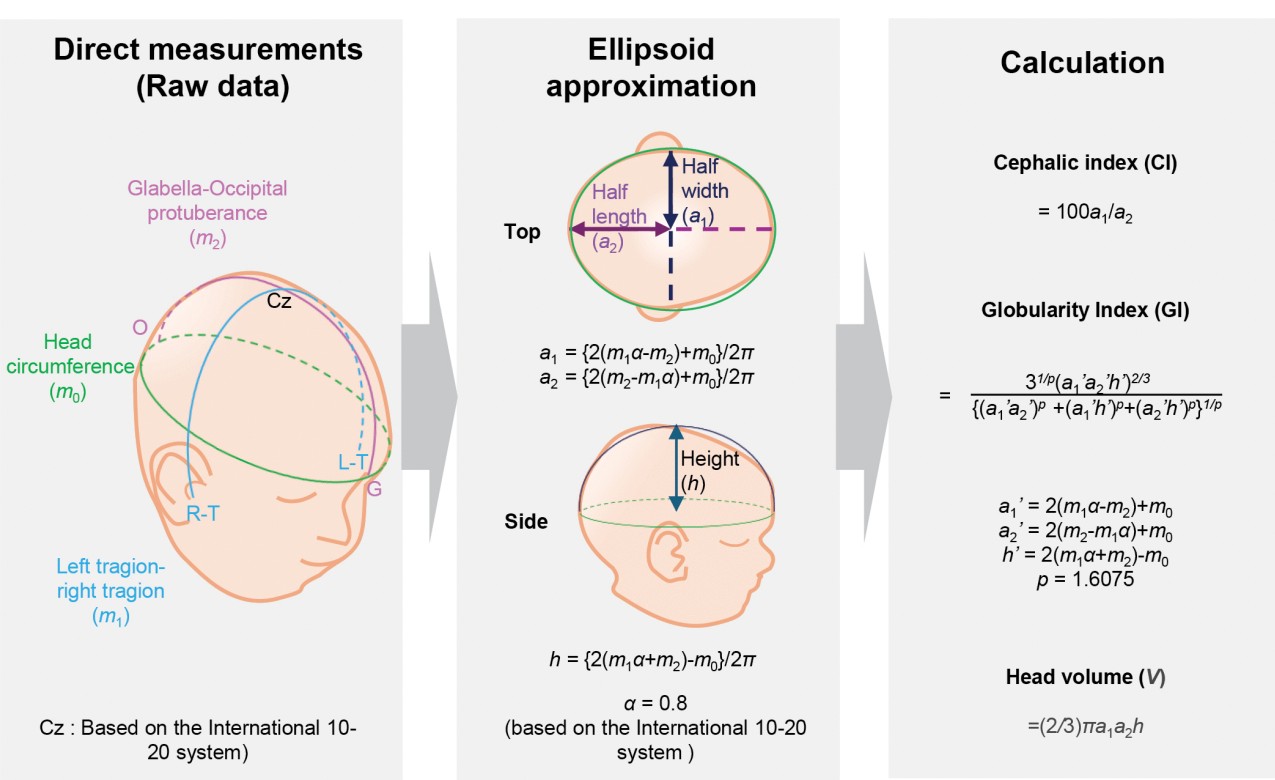

**Fig 1. Cephalic Index (CI), Globularity Index (GI) and head volume estimation.**

$m_0$, $m_1$, and $m_2$ were approximated as below using Gauss-Kummer equation. The over-the-head distance LT-RT was multiplied by α, which was set to 0.8, as the HC line intersects the LT-RT line near T3 and T4 of the international 10–20 system, at points corresponding to 10% of the total LT-RT distance away from LT and RT [23].

$$m_0 \simeq \pi(a_1 + a_2) \tag{7}$$

$$m_1 \alpha \simeq \frac{\pi(a_1 + h)}{2} \tag{8}$$

$$m_2 \simeq \frac{\pi(a_2 + h)}{2} \tag{9}$$

By solving the system of equations (1)-(3), $a_1$, $a_2$, and $h$ can be expressed using only $m_0$, $m_1$ and $m_2$ as shown below:

$$a_1 = \frac{\{2(m_1\alpha - m_2) + m_0\}}{2\pi} \tag{10}$$

$$a_2 = \frac{\{2(m_2 - m_1\alpha) + m_0\}}{2\pi} \tag{11}$$

$$h = \frac{\{2(m_1\alpha + m_2) - m_0\}}{2\pi} \tag{12}$$

Since cephalic index is the ratio between width ($2a_1$) and length ($2a_2$) in percentage, it can be expressed as follows:

$$\text{Cephalic Index (CI)} = 100 \times \frac{a_1}{a_2} \tag{13}$$

Head volume was assumed to resemble a half ellipsoid and obtained using $a_1$, $a_2$, and $h$ as shown below:

$$\text{Head volume} = \frac{2\pi a_1 a_2 h}{3} \tag{14}$$

## Globularity Index (GI) development

In addition to CI and head volume, Globularity Index (GI) was developed to assess how close the head shape was to a sphere. GI was defined as the ratio of the surface area of an ellipsoid $S_1$, estimated from the head shape measurements, to the surface area of a sphere $S_2$ with the same volume as the ellipsoid, as shown in Fig 2. To calculate this, the first step was to determine the radius $r$ of the sphere with the same volume as the ellipsoid using the following equation.

$$\frac{4\pi a_1 a_2 h}{3} = \frac{4\pi r^3}{3} \tag{15}$$

Then, $r$ was determined by the following equation.

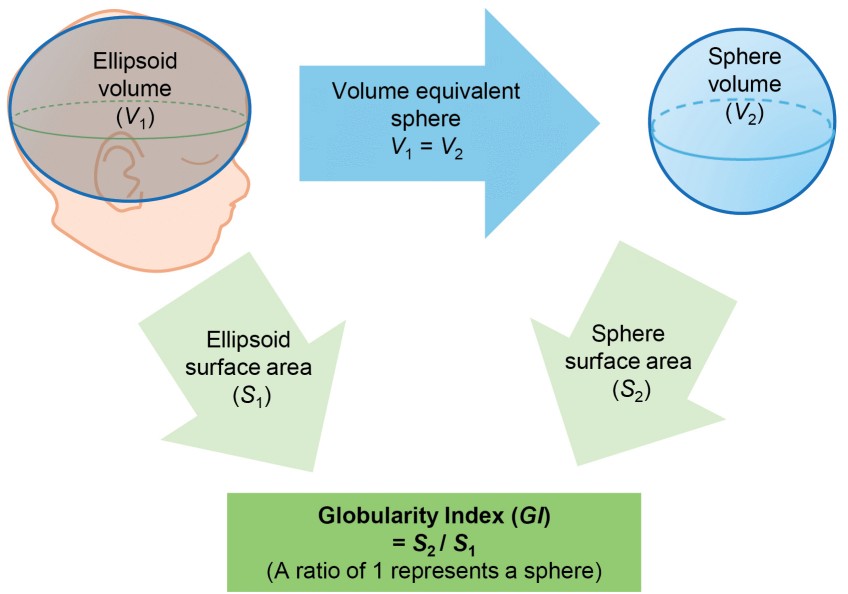

**Fig 2. A pictorial explanation of the Globularity Index (GI).**

$$r^3 = a_1 a_2 h \tag{16}$$

The surface area of an ellipsoid $S_1$ can be obtained using Knud Thomsen formula as below.

$$S_1 = \frac{4\pi \left\{ (a_1 a_2)^p + (a_1 h)^p + (a_2 h)^p \right\}^{1/p}}{3^{1/p}} \tag{17}$$

The surface area of a volume-equivalent sphere $S_2$ is given as below.

$$S_2 = 4\pi r^2 \tag{18}$$

Finally, GI can be calculated as below.

$$GI = S_2/S_1 = \frac{3^{1/p}(a_1'a_2'h')^{2/3}}{\left\{ (a_1'a_2')^p + (a_1'h')^p + (a_2'h')^p \right\}^{1/p}} \tag{19}$$

where

$$a_1' = 2\left( m_1\alpha - m_2 \right) + m_0 \tag{20}$$

$$a_2' = 2\left( m_2 - m_1\alpha \right) + m_0 \tag{21}$$

$$h' = 2\left( m_1\alpha + m_2 \right) - m_0 \tag{22}$$

$$p = 1.6075$$

Since the minimum surface area is given by the volume-equivalent sphere, the maximum GI value will be 1.00 and the closer the GI to 1.00, the closer the shape of the ellipsoid (head shape) to a sphere. Prior to the analysis and visualization, Fisher's z-transformation was applied to GI values as the values were likely to cluster around 0.9 to1.0.

### Data analysis and visualization

To investigate how the age impacted head measurements during the first 12 months following birth considering the effects of other factors including sex and birth year, factorial analysis of ANOVA (analysis of variance), a general linear model with age in months, sex, and birth year as main effects was conducted. The selection of predictors and potential interaction terms was guided by model comparison metrics, specifically the coefficient of determination ($R^2$) and Akaike Information Criterion (AIC), with CI treated as the response variable [24]. Type II ANOVA was implemented to account for the unbalanced design without interactions [25,26]. Estimated marginal means (EMMs) were computed for each age group, while averaging over the levels of sex and birth year with *emmeans* package in R (version 4.5.0) [27]. Pairwise comparisons of the EMMs following the effect of age in months were performed employing Wald t-test and Benjamini–Hochberg false discovery rate (FDR) correction, which was applied for multiple testing. Violin and box plots were generated for each age group to display the distribution of the raw data, while the overlaid data points and bars represented EMMs and their 95% confidence intervals. The lines in the boxplots illustrated the median and quartiles for each age group.

For birth year comparisons, only 3-month-old infants were selected due to the substantial number of records available for this age group relative to other monthly segments and to ensure uniformity as difference in age could impact the shape. In addition, 3-month-old was the only monthly segment which could be extracted for all birth years. The number of infants in each birth year group is presented in Table 1(d). For evaluation of the birth year effect on the measurements, type II factorial ANOVA was conducted, with birth year and sex as main effects. For the model selection, $R^2$ and AIC metrics were employed as mentioned above. This was followed by pairwise comparisons of EMMs for the birth year effect, adjusting for sex, with Wald t-test and FDR correction. Effect sizes are presented as absolute values in the results.

All statistical analyses and visualization steps were completed using R package *data.table, car*, *emmeans, lubridate* and *ggplot2*. (Supplementary Scripts S3 File)

## Results

### Alterations in head shape that occur over the initial 12 months following birth

**Direct measurements (left tragion-right tragion (LT-RT), glabella-occipital protuberance (G-O), and head circumference (HC)) and estimated measurements of half-width ($a_1$), half-length ($a_2$), and height ($h$).** Violin and boxplots describing the raw distribution of direct measurements (LT-RT, G-O, and HC) across age groups (2, 3, 6, and 12 months) along with overlaid EMM and 95% confidence level as error bars are displayed in Fig 3a-3c. Factorial ANOVA with age in months, birth year, and sex as main effects demonstrated that each factor had a significant effect across all direct measurements (for LT-RT, age: $F(3, 895) = 215.55$, $p < 0.001$, birth year: $F(9, 895) = 7.43$, $p < 0.001$, sex: $F(1,895) = 101.88$, $p < 0.001$; for G-O, age: $F(3, 895) = 90.18$, $p < 0.001$, birth year: $F(9, 895) = 4.05$, $p < 0.001$, sex: $F(1, 895) = 34.23$, $p < 0.001$; for HC, age: $F(3, 895) = 248.05$, $p < 0.001$, birth year: $F(9, 895) = 2.29$, $p = 0.02$, sex: $F(1, 895) = 166.67$, $p < 0.001$) (S1 Table in S1 File).

Across all direct measurements, a steady increase was observed as age progressed, averaging over the levels of sex and birth year. In all direct measurements, pairwise comparisons of EMMs following the main effect of age in months showed statistically significant differences between all pairs of age groups. This suggested that all direct measurements showed an overall increase from 2 to 12 months. EMMs for each age group, pairwise EMM comparisons and effect sizes

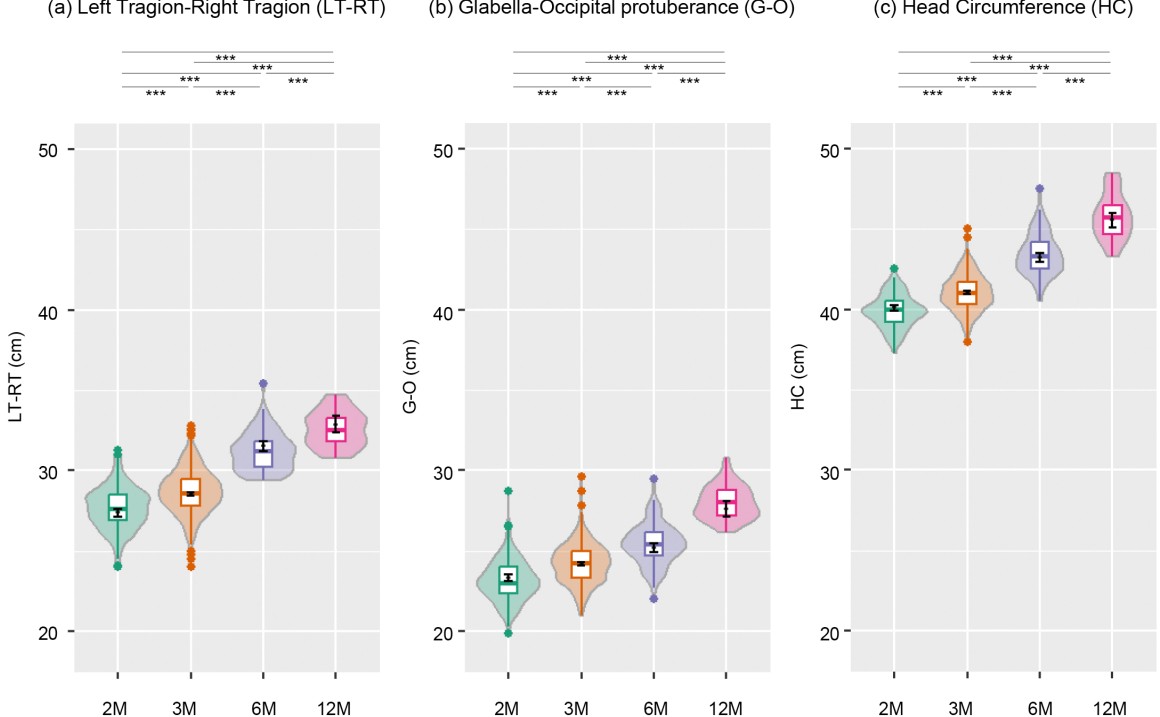

**Fig 3. Trends observed in the direct measurements (left tragion-right tragion (LT-RT), glabella-occipital protuberance (G-O), and head circumference (HC)) over the first 12 months after birth.** (a) LT-RT, (b) G-O, and (c) HC. Statistical differences across age groups (n = 207 for 2 months, n = 573 for 3 months, n = 100 for 6 months, and n = 29 for 12 months) were confirmed using factorial ANOVA with age group, sex and birth year as main effects, with the effect of age group on all direct measurements giving $p < 0.001$. The significance of pairwise differences (with lines connecting the pairs) between estimated marginal mean based on t-test and FDR correction are indicated as follows with asterisks: ***($p < 0.001$). For more details on post hoc analysis, see S2-S4 Table in S1 File.

for the direct measurements are presented in detail in S2-S4 Tables in S1 File. The original data plotted against days from birth are shown in S1 Fig in S2 File.

Fig 4a-4c depicts violin and box plots for the raw distributions of half-width ($a_1$), and half-length ($a_2$) and height($h$), at 2, 3, 6, and 12 months after birth, determined using the ellipsoidal approximation. Factorial ANOVA showed a significant effect exerted by each factor across all estimated measurements of $a_1$, $a_2$ and $h$, (for $a_1$, age: $F(3, 895) = 76.76$, $p < 0.001$, birth year: $F(9, 895) = 7.80$, $p < 0.001$, sex: $F(1, 895) = 47.73$, $p < 0.001$; for $a_2$, age: $F(3, 895) = 20.15$, $p < 0.001$, birth year: $F(9, 895) = 11.21$, $p < 0.001$, sex: $F(1, 895) = 5.78$, $p = 0.02$, for $h$, age: $F(3, 895) = 122.06$, $p < 0.001$, birth year: $F(9, 895) = 4.06$, $p < 0.001$, sex: $F(1, 895) = 43.30$, $p < 0.001$). (S5 Table (a)-(c) in S1 File).

Each estimated measurement presented a unique trajectory during the first 12 months. In reference to half-width ($a_1$), considering the main effect of age in months adjusted for birth year and sex, an increase from 2 months (5.93, 95% CI [5.85, 6.01]) to 3 months (6.10, 95% CI [6.06, 6.14]) was observed, followed by another increase from 3 to 6 months (6.88, 95% CI [,6.77, 6.99]) and no difference from 6 to 12 months (6.83, 95% CI [6.64, 7.02]) (Fig 4a, S6 Table(a) in S1 File). Pairwise comparisons revealed a statistically significant difference between all pairs of age groups ($p < 0.001$) except between 6 and 12 months for $a_1$, with a small effect size for 2 months versus 3 months ($d = 0.38$) and a large effect size for 3 months versus 6 months ($d = 1.75$). Details of mean differences, pairwise comparisons and effect sizes for $a_1$ can be found in S6 Table in S1 File.

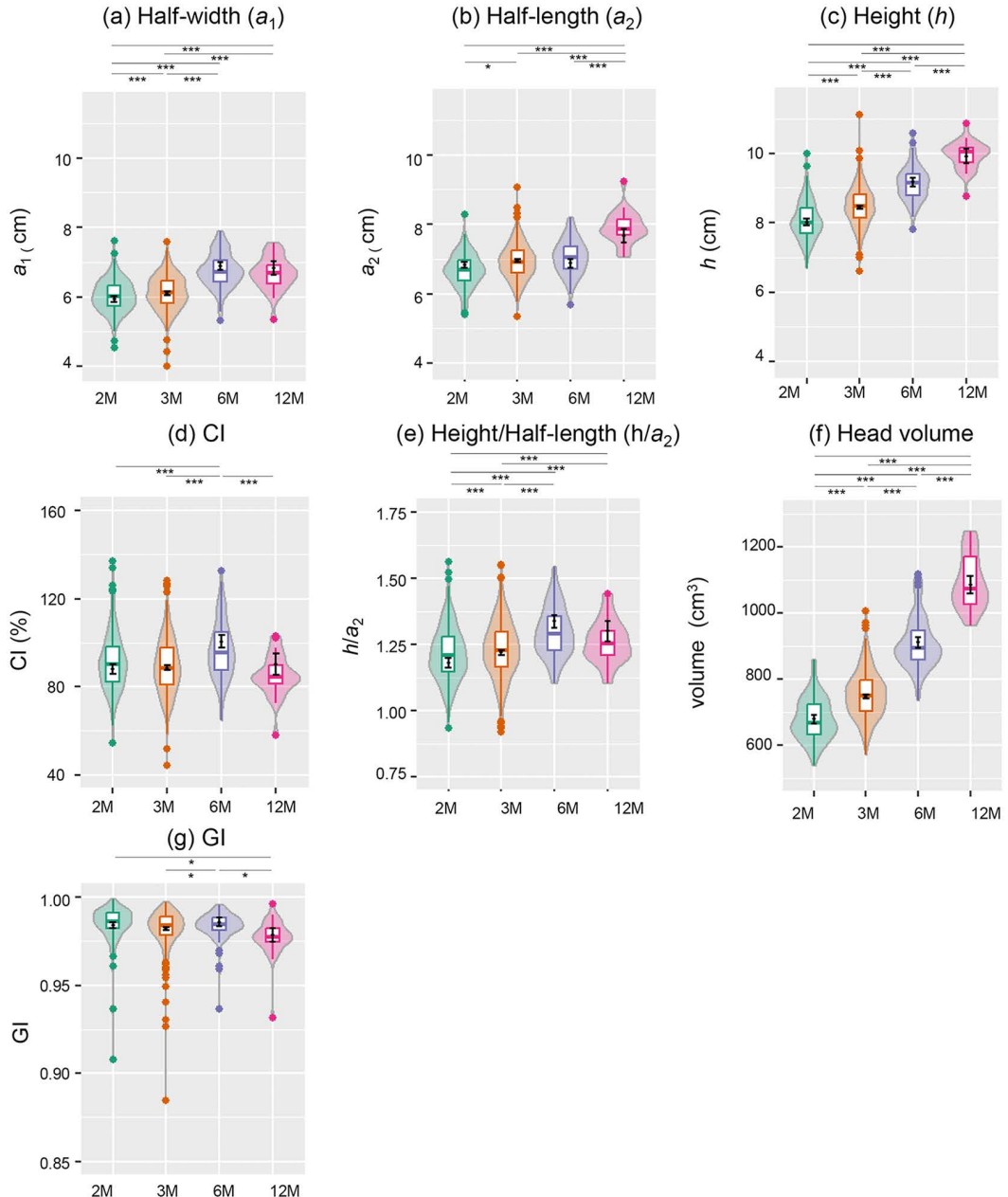

**Fig 4. Trends observed in the estimated measurements (half-width ($a_1$), half-length ($a_2$), height ($h$), CI, height/half-length ratio ($h/a_2$), head volume, and GI) over the first 12 months after birth.** (a) $a_1$, (b) $a_2$, (c) $h$, (d) CI, (e) $h/a_2$, (f) head volume, and (g) GI. Differences among the monthly segments (2 months: n = 207, 3 months: n = 573, 6 months: n = 100, 12 months: n = 29) were statistically significant, as determined using a factorial ANOVA with age in months, sex and birth year as main effects; the effect of age on all estimated measurements adjusting for sex and birth year was significant, with $p < 0.001$ for all measurements except GI ($p = 0.004$). The significance of pairwise differences between estimated marginal mean (with lines connecting the pairs) based on t-test and FDR correction are indicated with asterisks as follows: *($p < 0.05$), ***($p < 0.001$) and none (not significant). See S6-S12 Tables in S1 File for more details on post hoc analysis.

With respect to half-length ($a_2$), the EMM showed an increase from 2 months (6.83, 95% CI [6.74, 6.91]) to 3 months (6.96, 95% CI [6.91, 7.00]), succeeded by no change from 3 to 6 months (6.87, 95% CI [6.76, 6.99]) and further increase from 6 to 12 months (7.67, 95% CI [7.48, 7.86]) (Fig 4b, S7 Table(a) in S1 File). Pairwise comparisons presented statistically significant differences for all pairs except for the contrast between 2 and 6 months and between 3 and 6 months. 2 months was distinct from 3 months with a small effect size ($d = 0.28$, $p = 0.02$) and 6 months differed from 12 months with a large effect size ($d = 1.72$, $p < 0.001$). Details of mean differences, pairwise comparisons and effect sizes can be found in S7 Table in S1 File.

In reference to $h$, a constant increase in EMM was observed from 2 months (8.02, 95% CI [7.92, 8.11]) to 12 months (9.91, 95% CI [9.71, 10.12]) (Fig 4c, S8 Table(a) in S1 File). Pairwise comparisons showed statistically significant differences across all pairs with $p < 0.001$ and large effect sizes with absolute Cohen's $d$ ranging from 0.84 to 1.46 between adjacent age groups. Details of mean differences, pairwise comparisons and effect sizes for $h$ are included in S8 Table in S1 File.

In summary, these results indicated that there was a statistically significant increase in $a_1$ from 2 to 6 months, although to a different degree from 2 to 3 months and 3–6 months, with no significant difference observed between 6 and 12 months, and with reference to $h$, there was a constant increase across months. $a_2$ presented a unique pattern of a rise from 2 to 3 months, followed by a plateau from 3 to 6 months, and a further increase from 6 to 12 months reaching the highest value at 12 months. Scatter plots of the raw data points for $a_1$, $a_2$ and $h$ measurements can be found in S2 Fig in S2 File.

**Estimated measurements of CI, height-length ratio and head volume**

For CI, height-half length $(h/a_2)$ ratio, and head volume, factorial ANOVA incorporating age in months, sex and birth year as main effects indicated each factor exerted a statistically significant impact on all of the measurements, with the exception of birth year effect on volume (for CI, age: $F(3, 895) = 17.97$, $p < 0.001$, birth year: $F(9, 895) = 9.64$, $p < 0.001$, sex: $F(1, 895) = 7.56$, $p = 0.006$; for height-half length $(h/a_2)$ ratio, age: $F(3, 895) = 36.90$, $p < 0.001$, birth year: $F(9, 895) = 12.64$, $p < 0.001$, sex: $F(1, 895) = 8.95$, $p = 0.003$; for volume, age: $F(3, 895) = 353.34$, $p < 0.001$, birth year: $F(9, 895) = 1.35$, $p = 0.21$, sex: $F(1, 895) = 153.59$, $p < 0.001$). (Fig 4d-4f, S5 Table (d)-(f) in S1 File)

Given the changes displayed by $a_1$ and $a_2$, the CI, which represents the percentage of half-width($a_1$) to half-length($a_2$), demonstrated an upward shift in EMM from 3 months (88.53, 95% CI [87.39, 89.67]) to 6 months (100.57, 95% CI[97.64, 103.50]) and a subsequent drop from 6 to 12 months (90.12, 95% CI [85.22, 95.02]) (Fig 4d, S9 Table(a) in S1 File). The pairwise comparisons indicated that there was a significant difference with a large effect size between 3 and 6 months ($d = 1.02$, $p < 0.001$) and between 6 and 12 months ($d = 0.88$, $p < 0.001$). In addition, the level achieved at 12 months was not significantly different from 2 or 3 months, and there was no difference between 2 and 3 months. These results implied that the highest CI was achieved at 6 months among the age groups, and the remaining age groups exhibited similar CI. In other words, at 6 months, the head shape exhibited a tendency toward brachycephaly, which is the flattening of the head with or without vertical height [28], then returning to its original elongated form by 12 months, although the exact timing of the trend reversal remains to be unclear. Further information on mean differences as well as effect sizes can be found in S9 Table in S1 File.

With regards to height($h$)-to-half-length($a_2$) ratio in Fig 4e, and S10 Table in S1 File, a progressive increase in EMM from 2 months (1.18, 95% CI [1.16, 1.20]) to 6 months (1.34, 95% CI [1.31, 1.36]) and a subsequent period of no significant change were noted. Pairwise comparisons demonstrated that the ratio was smallest across the age groups at 2 months. 2 months differed from 3 months with a moderate effect size ($d = 0.41$, $p < 0.001$), and 3 months varied from 6 months with a large effect size ($d = 1.21$, $p < 0.001$), but 6 months and 12 months were statistically indistinguishable. These results symbolized that the height was highest relative to the half-length at 6 months among the earlier months, although it remained similar thereafter. More details can be found in S10 Table in S1 File.

As for head volume, a continuous increase was observed over the 12 months period as shown in Fig 4f, with statistically significant differences between all pairs ($p < 0.001$). Further details on mean differences, pairwise comparisons and effect sizes are presented in S11 Table in S1 File. The scatter plots of the raw distributions for CI, height/half-length ratio, head volume are shown in S3 Fig (a)-(c) in S2 File.

**GI**

Meanwhile, statistical analysis demonstrated a significant effect of age and birth year on the measurement of GI, but sex had no significant effect (age: $F(3, 895) = 4.49$, $p = 0.004$, birth year: $F(9, 895) = 4.25$, $p < 0.001$, sex: $F(1, 895) = 0.23$, $p = 0.63$) (S5 Table (g) in S1 File).

EMMs obtained for GI averaged across birth year and sex presented a pattern resembling that observed in CI, specifically an increase from 3 month to 6 months followed by a decrease from 6 months to 12 months as shown in Fig 4g and S12 Table in S1 File. Pairwise comparisons revealed that the GI at 2 months was higher than 12 months ($d = 0.58$, $p = 0.03$), but the difference was unclear in contrast with 3 months and 6 months. In addition, the GI increased significantly from 3 to 6 months ($d = 0.40$, $p = 0.02$) and decreased from 6 to 12 months ($d = 0.79$, $p = 0.01$) but the difference between 3 and 12 months was not significant.

The statistically significant increase from 3 to 6 months with an ensuing decrease from 6 to 12 months in GI denoted that the shape became more spherical at 6 months but diverted back to an elongated form by 12 months. This coincided with the tendency toward brachycephaly at 6 months with subsequent reversal by 12 months in CI. However, whether there was a decrease in GI from 2 to 3 months or the level attained at 12 months was lower than that at 3 months remained uncertain. EMMs, pairwise comparisons, and effect sizes for GI are included in S12 Table in S1 File and scatter plots are displayed in S3 Fig (d) in S2 File.

**Sex differences across age**

In relation to sex differences across different age groups, as mentioned earlier, statistical analysis suggested that the impact of sex was significant across all direct and estimated measurements apart from GI, adjusting for age in months and birth year. The differences between males and females were uniform across age groups as the linear model included main effects of age, sex, and birth year with no interactions and the effect of sex was assumed to be additive and constant across the levels of age and birth year. Both males and females presented changes over time that were consistent with the trends observed in the overall samples, with females displaying significantly lower values as described in S4-S5 Figs in S2 File and S13-S22 Tables in S1 File.

**Differences observed across birth years from 2010–2019**

To elucidate the contribution of birth year on the head measurements, 3 months group was selected given the sizable number of available records compared with other age groups and the feasibility of extracting this age group across all birth years, as infants at a particular stage of development were recruited during the specific period, and there were biases in the age distribution across the birth years as described in Fig 5.

Factorial ANOVA with birth year and sex as main effects displayed that birth year had a significant effect across all estimated measurements except head volume while sex exerted a significant effect only on $a_1$, $a_2$, $h$ and volume (for $a_1$, birth year: $F(9, 562) = 9.44$, $p < 0.001$, sex: $F(1, 562) = 19.19$, $p < 0.001$; for $a_2$, birth year: $F(9, 562) = 13.75$, $p < 0.001$, sex: $F(1, 562) = 6.79$, $p = 0.009$; or $h$, birth year: $F(9, 562) = 3.97$, $p < 0.001$, sex: $F(1, 562) = 26.71$, $p < 0.001$; for CI, birth year: $F(9, 562) = 12.60$, $p < 0.001$, sex: $F(1, 562) = 1.46$, $p = 0.23$; for GI, birth year: $F(9, 562) = 2.78$, $p = 0.003$, sex: $F(1, 562) = 0.17$, $p = 0.68$; for height-length ratio ($h/a_2$), birth year: $F(9, 562) = 16.70$, $p < 0.001$, sex: $F(1, 562) = 3.41$, $p = 0.07$; for volume, birth year: $F(9, 562) = 1.41$, $p = 0.18$, sex: $F(1, 562) = 84.93$, $p < 0.001$) (S23 Table in S1 File).

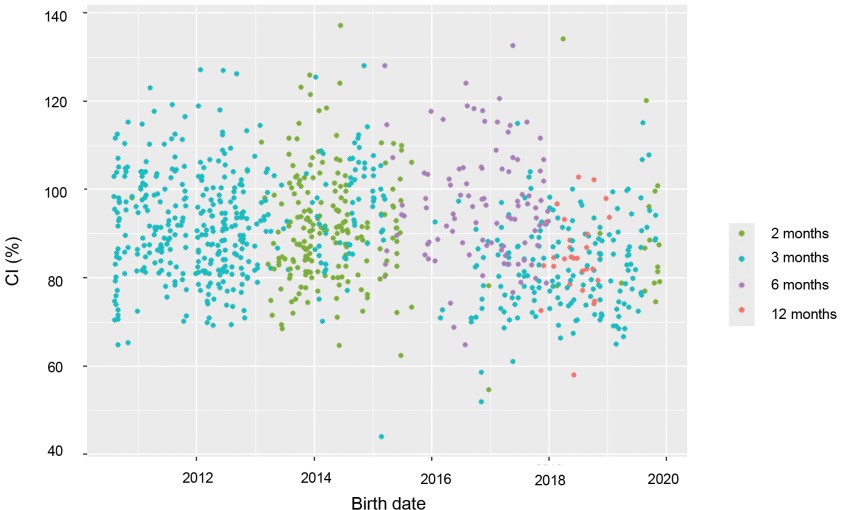

**Fig 5. Scatter plot of CI versus birth year.**

Violin and boxplots describing the distribution of raw data along with overlaid EMM and 95% confidence level as error bars across birth years are displayed in Figs 6, 7. In the figures, a letter was allocated to each birth year group in a sequential manner from left to right, "a" to "j". When the statistically significant difference was identified between any of the birth year pairs according to the pairwise post hoc test, the letter designated to the group that came first in the sequence was added above the violin plot of the group that appear later in the sequence. The difference between the birth years sharing letters was significant at either p < 0.05 (in black), p < 0.01 (in blue), or p < 0.001 (in red) level.

There was a notable difference in the range of EMM for $a_1$ and $a_2$ between the birth years preceding and following 2016. The EMM for $a_1$ observed during the period from 2010 to 2015 was likely higher than that from 2016 to 2019, whereas $a_2$ presented the opposite trend, being lower during the same period. Meanwhile, $h$ displayed no consistent trend, there was no clear differentiation of the first six years from the following years.

On one hand, pairwise comparisons revealed that the period from 2010 to 2015 was significantly different from 2016 to 2019 for $a_1$ and $a_2$, with no or few pairs showing differences among 2010–2015 themselves, as shown in S24-S29 Tables in S1 File and Fig 6a-6b.

With reference to $a_1$, each of the years in the earlier period 2010–2015 exhibited a significantly higher value relative to each year in the later period 2016–2019, with all pairwise cross-period (2010–2015 vs 2016–2019) comparisons showing moderate to large effect sizes ($d = 0.54$–$1.33$, $p < 0.05$) (Fig 6a, S24-S26 Tables in S1 File). For $a_1$, no significant within-period variations were observed.

On contrary, regarding $a_2$, each year in the 2011−2015 range showed a lower value than all years in the 2016−2019 range, with moderate to large effect sizes ($d = 0.56$–$1.75$, $p < 0.05$) (Fig 6b, S27-S29 Tables in S1 File). Additionally, 2010 displayed lower $a_2$ in contrast to 2016 ($d = 1.01$, $p < 0.001$) and 2018 ($d = 0.56$, $p = 0.01$). Furthermore, there were statistically significant within-period heterogeneity among the earlier years. A higher $a_2$ was observed in 2010 than in 2011 and 2013−2014 ($d = 0.54$–$0.74$, $p < 0.05$), a lower value was observed in 2011 than in 2012 ($d = 0.29$, $p = 0.04$), and a higher value was noted in 2012 than in 2014 ($d = 0.48$, $p = 0.006$). Within the later years, the only difference observed was between 2016 and 2019 ($d = 0.71$, $p = 0.02$), with 2016 displaying a higher value.

On the other hand, the post-hoc test revealed significant differences between only a fraction of birth year categories for $h$. (S30-S32 Table in S1 File, Fig 6c) With regards to $h$, differences were observed within the first six years between 2010

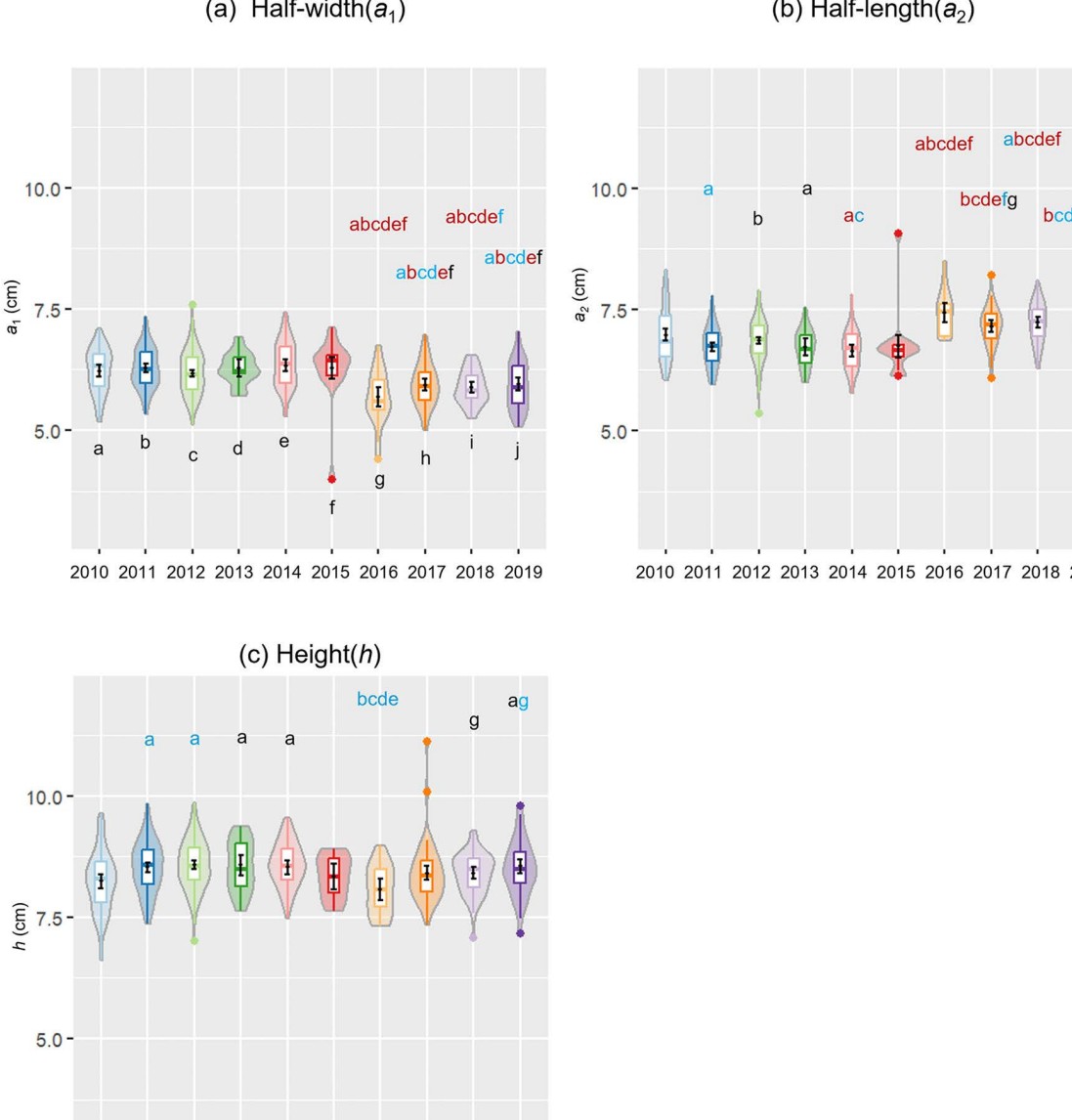

**Fig 6. Trends observed in the estimated measurements (half-width ($a_1$), half-length ($a_2$) and height ($h$)) at 3 months over the period from 2010 to 2019.** (a) $a_1$, (b) $a_2$, and (c) $h$. A letter was allocated to each birth year from left to right as indicated inside the box. The letter assigned to the group that came first in the sequence was added above the violin and box plots of the group that appear later in the sequence if the difference was significant at either $p < 0.05$ (in black), $p < 0.01$ (in blue), or $p < 0.001$ (in red) level based on the pairwise t-test with FDR correction conducted to contrast estimated marginal means. The number of infants was n = 52 for 2010, n = 100 for 2011, n = 148 for 2012, n = 25 for 2013, n = 52 for 2014, n = 15 for 2015, n = 19 for 2016, n = 51 for 2017, n = 63 for 2018, and n = 48 for 2019. See S24-S32 Tables in S1 File for details of post hoc analyses.

and each year across a range from 2011 to 2014, driven by the lower value in 2010 ($d = 0.56$–$0.67$, $p < 0.05$). Additionally, there was heterogeneity within the succeeding four years. 2016 displayed a lower value than 2018 ($d = 0.66$, $p = 0.048$) and 2019 ($d = 0.93$, $p = 0.007$). The only intra-period disparities noted were between 2010 and 2019, with 2010 demonstrating a lower value ($d = 0.60$, $p = 0.02$) and between each year in the 2011–2014 range and 2016, with 2016 showing a

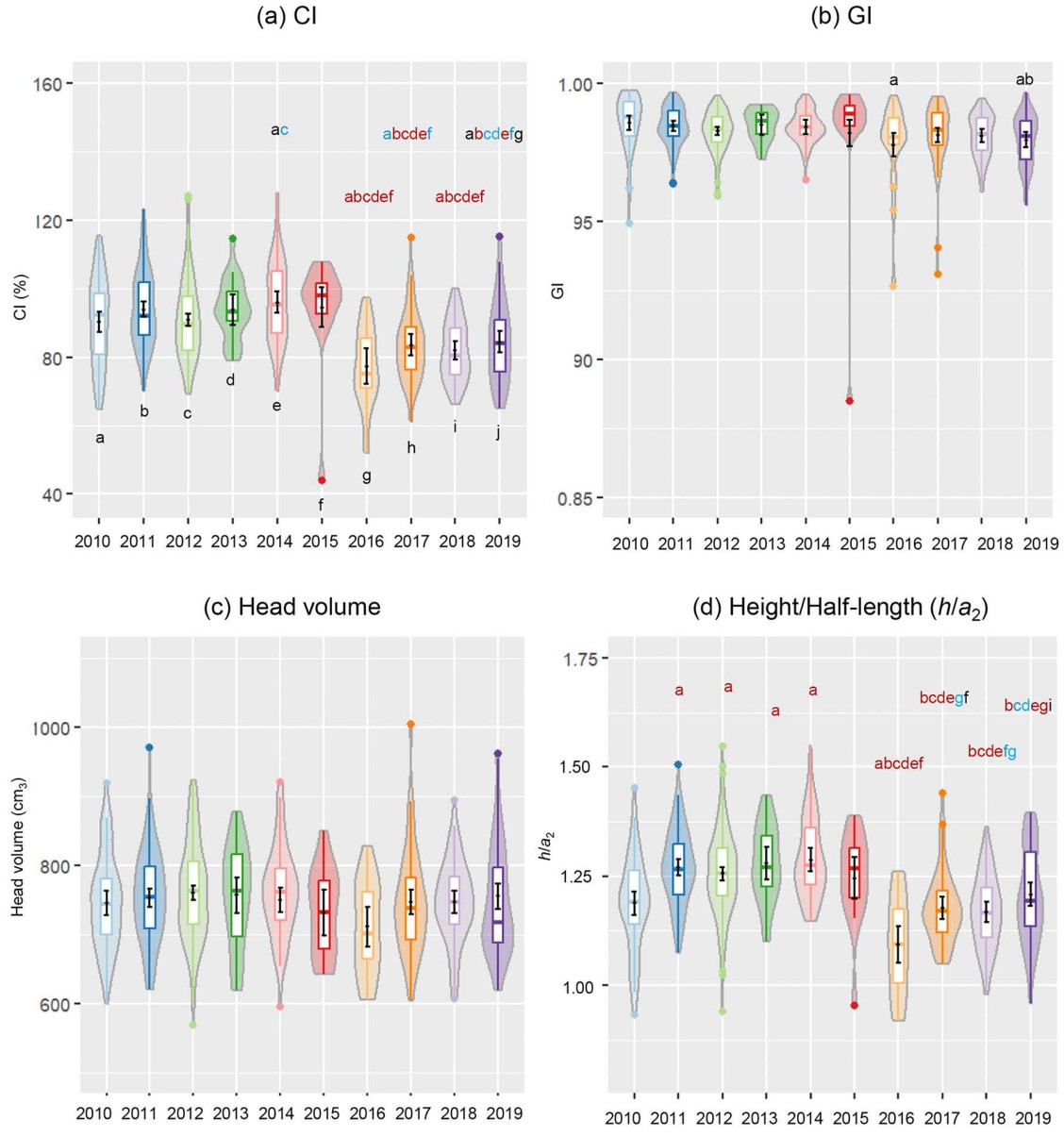

**Fig 7. Trends observed in the estimated measurements (CI, GI, head volume, and height/half-length ($h/a_2$)) at 3 months over the period from 2010 to 2019.** (a) CI, (b) GI, (c) head volume, and (d) $h/a_2$. A letter was allocated to each birth year from left to right as indicated inside the box. The letter assigned to the group that came first in the sequence was added above the boxplot of the group that appear later in the sequence if the difference between estimated marginal means was significant at either $p < 0.05$ (in black), $p < 0.01$ (in blue), or $p < 0.001$ (in red) level based on the pairwise post hoc t-test with FDR correction. 2010 (n = 52), 2011 (n = 100), 2012 (n = 148), 2013 (n = 25), 2014 (n = 52), 2015 (n = 15), 2016 (n = 19), 2017 (n = 51), 2018 (n = 63), and 2019 (n = 48). See S33-S44 Tables in S1 File for details of post hoc analyses.

lower value ($d$ = 0.89–1.00, $p < 0.01$). Raw data point plots for the direct measurements and estimated measurements ($a_1$, $a_2$, and $h$) can be found in S6-S7 Figs in S2 File.

Altogether, these suggested that the years spanning from 2010 to 2015 exhibited trends that likely differ from those in the following years for $a_1$ and $a_2$, with higher $a_1$ and lower $a_2$ than those observed in the subsequent years. In addition, $h$

presented a marked decrease in 2016, falling below the levels recorded in some of the previous years, although no directional temporal pattern was identified.

Due to the likely higher distribution of $a_1$ and the lower distribution of $a_2$ during the first six years compared to the subsequent years, the resulting CI values at 3 months were also inclined to be elevated during the first six years (2010–2015) as can be seen in Fig 7a and S33 Table in S1 File. In contrast, GI and head volume remained relatively constant (Fig 7b, 7c, S36 Table in S1 File, S39 Table in S1 File).

In reference to CI, post-hoc test confirmed that each of those in 2010–2015 range differed from those in 20162019 range significantly ($p < 0.05$) (Fig 7a, S33-S35 Tables in S1 File). The earlier years exhibited higher values with moderate to large effect sizes ($d = 0.51$–$1.65$, $p < 0.05$). Additionally, there was diversity within the years preceding and following 2016. Among the first six years, CI was higher in 2014 than in 2010 ($d = 0.51$, $p = 0.02$) and 2012 ($d = 0.46$, $p = 0.009$). Between the years following 2016, CI was lower in 2016 than in 2019 ($d = 0.63$, $p = 0.03$).

Relating to GI, post-hoc test revealed statistically significant variation only in a few of the pairs ($p < 0.05$). Specifically, GI in 2010 was higher than in both 2016 ($d = 0.83$,) and 2019 ($d = 0.64$) (Fig 7b, S36-S38 Tables in S1 File). In addition, GI in 2011 was higher than in 2019 ($d = 0.52$). As these results suggested, in terms of closeness to a sphere, there were no differences observed between the years before and after 2016.

Birth year had no statistically significant effect on volume as mentioned above, suggesting that the morphological changes did not accompany changes in volume. (Fig 7c, S39-S41 Tables in S1 File)

Height/half-length also showed a trend that resembled that observed in CI, with the post-hoc test displaying a considerable difference between each of those in the 20112014 range and those in the 20162018 range, with the former years demonstrating higher values ($d = 0.51$–$2.07$, $p < 0.01$) (Fig 7d, S42-S44 Tables in S1 File). In terms of disparities between the years from the earlier period (2010–2015) and the later period (2016–2019), additionally, height/half-length ratio in 2010 was higher than in 2016 ($d = 1.01$, $p < 0.001$) and the value in 2015 exceeded that observed in any of the years in 2016–2018 interval ($d = 0.72$–$1.61$, $p < 0.05$). Comparisons among the earlier years showed that the ratio in 2010 was lower than each of those in 2011–2014 span ($d = 0.72$–$1.06$, $p < 0.001$), but no other differences were observed. With regards to within-period differences detected among the later years, the ratio in 2016 was lower than in each of 2017–2019 period ($d = 0.80$–$1.22$, $p < 0.01$) and the ratio in 2018 was lower than in 2019 ($d = 0.42$, $p = 0.04$). The raw data point distributions for CI, GI, height/half-length and volume are included in S8 Fig in S2 File.

These outcomes together suggested that although the head volume and GI remained relatively stable over the years, there was a shift in the average head shape in more recent years, towards front-to-back elongation rather than flattening, with a lower height to half-length ratio.

## Discussion

In this study, we strived to elucidate the typical pattern of morphological development in the head during the first 12 months of life in healthy infants with a uniform ethnic background. To the best of our knowledge, there are a limited number of studies on the natural course of morphological development in the cranium in healthy infants. Although there are reports on the morphological changes in the cranium, many studies focused on comparisons between infants with and without cranial deformation or pre- and post-surgery [10,14,29–32]. Moreover, only a few studies capture the alterations that occur in shorter intervals during the first year or address variations across the subpopulations using a large database.

First, we identified the trends in the measurements over the first 12 months of life. Screening across all measurements, it was evident that all direct measurements, height ($h$) and volume exhibited an overall upward shift, as indicated by the increasing estimated marginal mean values. In reference to half-length ($a_2$), the value increased from 2 to 3 months, remained similar between 3–6 months, and reached its maximum at 12 months. Meanwhile, half-width ($a_1$), height/half-length ratio and the CI reached the maximum value at 6 months, followed by a subsequent drop or

no change by 12 months. This implied that the head underwent front-to-back flattening with increasing vertical height at 6 months but reverted to an elongated form by12 months with no change in the height/half-length ratio, although the exact timing of the trend reversal remained uncertain. Consistent with these trends, the GI also displayed an elevation from 3 to 6 months with a subsequent drop by 12 months, which refers to a closer resemblance of the head shape to a sphere at 6 months compared to that at 3 or 12 months. However, it remained unclear at which age the GI peaked or reached its lowest across age groups. The analysis of the CI coupled with the GI illustrated that the head shape resembled a ball slightly stretched laterally (right-to-left) at 6 months and horizontally (front-to-back) at 12 months.

It is during the first few months that craniosynostosis, a congenital malformation characterized by the premature closure of cranial vault sutures, which may result in functional complications becomes more visually evident. In addition, deformational plagiocephaly, an acquired head deformity may occur [1,16,33]. Thus, it is imperative to understand the morphological developments of the normal brain especially in the first year of life.

## Developmental changes in CI during the first 12 months of life

With regards to the estimated measurements such as CI and volume, the direct comparisons of our results with that of the previous studies can be challenging as the landmarks used to compute these measurements in this study may differ from those in other studies, producing different linear dimensions [5]. It is also crucial to consider how each study segmented the age, how many infants were included in each age group and which scanning technique was used, for these factors may further contribute to the discrepancies, in addition to the characteristic variations.

Because the CI norms were largely established several decades ago, there were a few preceding studies that aimed to establish revised CI norms tailored for specific populations. In one example, Likus et al. (2014) sought to update CI standards for Polish children under the age of 3 years by analyzing cross-sectional CT scan images acquired from 180 children. The mean CI exhibited a modest increase during the first 12 months [13]. A more recent study by Phelan et al. (2021) also aimed to renew CI norms for American children using a large and diverse dataset containing measurements collected with manual calipers from 870 children in one of the five age categories (0–3 months, 3–6 months, 9–12 months, 2–3 years, and 12–14 years) [3]. In this context, CI peaked in the 3–6 months segment and declined in the older age groups, irrespective of ethnic variations.

In contrast to these studies that measured CI directly, several studies obtained and assessed cranial parameters including CI through 3D reconstructed CT images or 3D photographs. Myer-Marcotty et al. (2018) applied 3D photogrammetry to conduct a longitudinal analysis of cranial development involving healthy Caucasian infants, divided into four age groups (4, 6, 8, and 10 months) [34]. They evaluated CI and observed no significant difference between 4 and 6 months, although a subsequent gradual decline was noted. In another study, Meulstee et al. (2019) analyzed both 3D photographs and CT scan images obtained from infants between the ages of 0–54 months [35]. Although they observed no notable trend in CI, the growth heat maps generated by contrasting the average shape for each age group revealed a trend indicating growth in frontal and parietal regions from 3 to 6 months as opposed to just the anterior region from 6 to 12 months.

In another example, Liang et al. (2023) reconstructed 3D cranial images using CT scan images from 217 individuals and described the craniofacial growth over the first 48 months of life [15]. They derived multiple skull indices including CI and performed PCA and form space analysis to characterize the morphologies. They discovered that the shape evolved towards front-to-back flattening from 0 to 6 months, achieving the most flattened form in both males and females in the 3–6 months segment, then returning to a less flattened form in the 6–12 months segment.

As can be seen, earlier investigations conducted across multiple countries yielded varied results regarding CI. Nevertheless, some presented a commonality in that changes were likely to occur at the 6-month mark comparing the cranial parameters taken before and after this period, which agreed with our observations.

## Comparisons with Japanese studies

In relation to healthy Japanese infants, there are several prior studies. Many of these employed CT scan images, and those applying 3D scanner images only started to emerge in recent years [36,37]. These studies demonstrated trends in CI and 3D morphology that were mostly aligned with ours.

In one example, Koizumi et al. (2010) conducted a cross-sectional study utilizing CT scan images obtained from 104 children aged 0–3 years to establish updated CI standards for Japanese children [11]. They reported a steady increase in the CI until 7–9 months, which was followed by a sustained drop until 3 years of age.

Kuwahara et al. (2019, 2020) also used CT scan images to evaluate craniofacial features of healthy Japanese infants [8,38]. They reconstructed 3D craniofacial images applying homologous modelling and created group-specific average models. In one of their studies, they superimposed each age-specific average model on the model of the adjacent age group to analyse the area and the degree of change to define the healthy growth pattern. The analysis was performed on 120 infants aged between 1–17 months. During the period from 1 to 4–5 months, they observed growth in the whole cranium except for the occipital region, with the CI continuously increasing and reaching the highest value in the 4–5 months segment. This was followed by growth in the frontal and occipital region from 4–5 months to 6–8 months and in posterior temporal and occipital regions from 6–8 months to 12–17 months. The continuous expansion of the whole cranium seen during the earlier months in their study may explain our observation of front-to-back flattening. Furthermore, the subsequent development in the regions limited to frontal and occipital regions may also explain the increase in $a_2$ and not in $a_1$ in our study, resulting in lower CI after 12 months.

Although it remains uncertain how these morphological changes materialized in healthy infants, one of the contributing factors may be the growth of the specific regions of the cranium occurring at different points in time, as seen in prior studies [8,35]. Because the infant cranium is soft and flexible to accommodate a rapidly growing brain, especially during the first few months, understanding regional growth differences in the brain may help to explain the resulting changes in the head shape. Previous MRI studies of the infant brain reported that the exponential growth of gray matter is responsible for the volume increase during the first two years of life. This occurs along with the regionally heterogeneous and age specific expansion of the cortical surface area, which expands by 76% from birth to first year [39–41]. The major cortical folding patterns are known to be well established at term birth and preserved during the first year of life, with mostly tertiary folding rapidly developing after birth [42]. Though the exact timing is not yet clear, lateral frontal, lateral parietal and occipital regions are known to present faster growth in surface area during the initial two years [39,40,43,44]. Currently, the relationship between the morphologies of the cranium, brain and head remains to be understood in detail. Therefore, the advancement of new macroscopic quantification methods such as GI should help to add more knowledge and bridge the gaps between them.

Additionally, one of the external factors frequently discussed in relation to infant head shape is supine sleep position which promotes brachycephaly during the earlier months of life, but this often resolves as the infant becomes more capable of making movements and gains head control [3,45,46].

## Sex differences

In many of the existing studies, the sample size was often too small to allow grouping of the infants by monthly categories or other background characteristics, one of the reasons possibly being the usage of CT scanning, which can be difficult to conduct on healthy infants due to the associated risks and operational challenges.

In this study, we strived to identify how the variations in background characteristics may take part in discrepancies in the head measurements. Firstly, there were visible differences between sexes. Males were more likely to present higher distributions for the measurements excluding the GI although the trends were alike between sexes. However, prior studies exhibited varied results regarding sex differences.

With reference to CI values, several studies reported no major sex differences during specific parts of the first year, as well as throughout the entire year [6,13,34]. In terms of 3D craniofacial morphology, Liang's (2023) report indicated little

sign of sex differences, although the males appeared to be larger in sizes. Yet, to a much lesser extent, the relationship between head volume proportions and age, as well as head volume proportions and size, varied by sex in their study [15,47].

Relating to Japanese population, the results from recent studies were comparable to ours, with males presenting higher CI values and volumes during the first year, although one study reported varied results [7,11,36,37].

### Birth year differences

As our statistical analyses indicated, one of the characteristics that influenced the indices the most was birth year. Thus, violin and box plots as well as estimated marginal means of CI, GI, height/half-length and volume at 3 months were displayed by birth year to visualize trends (Fig 7). During most of the first five years CI and height/half-length showed estimated marginal mean values that were likely to be higher compared to the following four years. With respect to CI, pairwise comparisons revealed that each year in the 2010–2015 range consistently differed significantly from the years in the 20162019 range. Additionally, there was heterogeneity within earlier years and later years. In respect of height/half-length, significant differences were observed between 2010 and each year in the period from 2011 to 2014, and between 2018 and 2019, in terms of within-period heterogeneity. Regarding the inter-period differences, a higher value was observed in 2010 than in 2016, the years in the 2011–201 range were significantly different from those in the 20162019 range, and 2015 was significantly different from 2016–2018. However, as for the GI, there was a pronounced difference only between 2010 and 2016, between 2010 and 2019, and between 2011 and 2019. Note that, as for the head volumes, there were no significant differences between the birth years.

These demonstrated that the 3 months olds with recent birth years were likely to present a more front-to-back elongated head with a lower height to half-length ratio in contrast to those with previous birth years. In addition, this may imply that the morphological standards for healthy infants in the first year of life need to be updated at least once or twice in a decade.

Although existing studies have reported secular changes in relation to the cranial shape in adults, their causes remain unclear. In addition, trends seem to vary depending on the country in question. In Japan and Korea specifically, a persistent increase in CI was observed in adults born in years leading up to the 1960s or 1970s, and those born after the 1970s experienced a reversal of this trend, exhibiting a continuous decline in CI [20,48,49]. This debrachycephalization, which commenced in the 1970s and continued until recent years, was also seen in children aged 3–5 years [20]. Hence, our results may indicate that similar secular changes occurred in infants in the first year of life. To our knowledge, no previous studies mentioned the secular changes occurring in 3-month-old infants, which makes this research the first to report this in detail.

### HC comparison with the data from Japanese national growth survey on preschool children

Additionally, to confirm that the HC measurements obtained in this study were proportional to the previously reported data from the Japanese national growth survey on preschool children, HC measurements from all records including those without the LT-RT and G-O measurements and/or belong to age groups other than 2, 3, 6, or 12 months were plotted against the age to construct a scatter plot and contrasted with the Japanese growth survey data published for the period from 1960 to 2010 in Fig 8 (Boxplots and the scatter plot for 2, 3, 6 and 12 months only but including those without LT-RT and G-O can also be found in S9 Fig in S2 File). Moreover, a logarithmic approximation curve of the scatter plot was added. To match with the age range of the growth survey data, infants older than 375 days were excluded from the plot. As the growth survey data only provided the average HC measurement for the corresponding age range and sex, for example 37.47 cm in males and 37.0 cm in females for the age range 2–3 months, we first summed the male average and the female average and divided by two to obtain the mean for the combined population [50]. Following this, we plotted the combined mean against the midpoint of the age range; for example, for the age range 2–3 months (equal to or older than

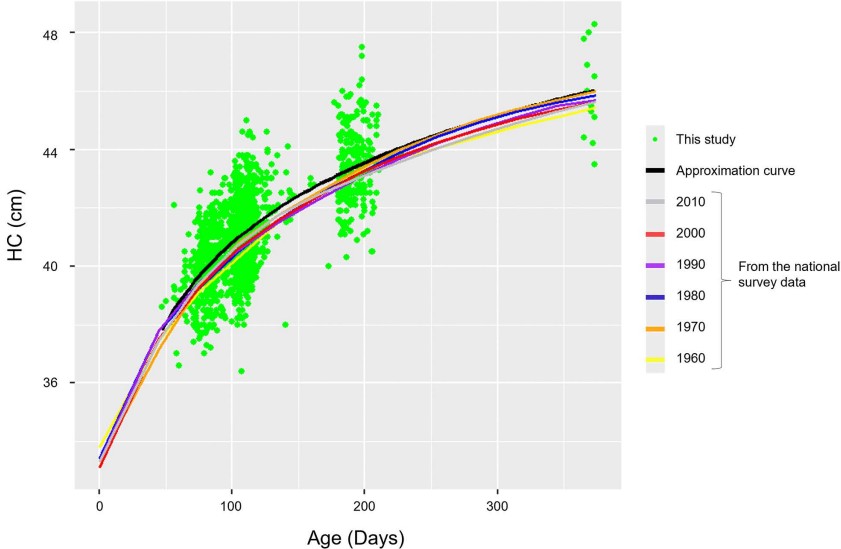

**Fig 8. Comparative review of head circumference (HC) measurements from this study and the national growth survey on preschool children for 1960 to 2010.** A scatter plot of all HC measurements (dots in green) including infants whose age did not correspond to 2, 3, 6 or 12 months but removing infants older than 375 days to match the age range of the national growth survey data (n = 1,958). A logarithmic approximation curve (a curve in black) was overlayed with government data (lines in colors) made available for the period from 1960 to 2010. As the national growth survey data only provided the average measurement for the corresponding age range and sex, the mean values combining both sexes were plotted against the midpoints of the age range.

2 months but younger than 3 months), the mean was plotted at 74.5 days. Our data demonstrated a trend which overlapped with that of the growth survey data, as can be seen from the approximation curve. When compared to the earlier studies that applied 3D photographs, the average HC measurements in our study were mostly identical accounting for age and sex [7,35–37]. These suggested that our HC data, though obtained using tape measures, were in alignment with the trends of the established standards and can be a reliable source for investigating the Japanese infant head shape.

### Existing methodologies and the necessity of additional 3D evaluation method

Infant cranial morphology can be captured and assessed through a range of methodologies involving tape measures, calipers, CT scans and 3D photogrammetric scanners. Among these, tape measures and calipers are non-invasive, easy to apply and affordable, thus making these the most widespread in clinical practice. However, these manual measures are usually limited by their inability to provide details regarding the 3D morphology. Conversely, CT scans can be used to precisely reconstruct the 3D structure but can be disadvantageous to healthy infants because of radiation exposure. Therefore, it may not be suitable for frequent and long-term monitoring. In the last few years, significant progress has been made in the application of 3D stereophotogrammetry, successfully conquering the challenges associated with traditional methodologies. Nonetheless, it may still necessitate the use of additional costly instruments.

Under these circumstances, it is beneficial to develop a method which allows non-invasive, quick and inexpensive assessment and long-term monitoring of healthy infant head shape in 3D. Extraction of additional objective 3D information including head shape roundness and height from readily available tape measurements should be useful, as in some cases physicians rely on subjective evaluation such as appearance for the assessment of morphological improvements after medical or non-medical interventions, due to lack of well-established methods other than two-dimensional CI [7]. For these reasons, we developed an alternative method for 3D quantification of the head shape which can be conducted

using the data that are already available in day to day clinical and non-clinical settings or easily collected, such as over-the-head distances and circumferences. This alternative method, GI, aims to quantify the extent to which the head shape is deviating from a sphere and add information to supplement CI. Together with CI, GI allows intuitive understanding of the changes in the head shape, although further investigations are needed for validations of GI use as this is still an exploratory phase.

In a recent study, authors applied spherical harmonics to model the distances from the actual head to an ideally fitted ellipsoid using data obtained with a 3D photogrammetric system, based on the assumption that the ideal head shape is an ellipsoid [51]. The study investigated the diagnostic performance of spherical harmonics for differentiating infants with deformational plagiocephaly from those without and found that their method outperformed conventional asymmetry indexes.

In contrast, the present study characterizes population-level developmental changes in head roundness rather than detecting the deviation from the ideal reference shape. Although in our study GI was derived by dividing the ellipsoid surface area by the volume equivalent sphere surface area, this approach can also be applied to more complex head shapes than an ellipsoid. Since the minimal surface area is achieved by a sphere and deformation of a sphere in any direction, including local surface undulations, leads to an increase in surface area, the GI defined by surface ratios serves as a comprehensive indicator of deviation from sphericity. Using minimal and relatively easy measurements to quantify morphological characteristics has also enabled the collection of data from many infants across a wide range of ages in months. To quantify asymmetry in specific directions or deviations from a typical morphology, more detailed three-dimensional measurements and the establishment of age-related changes in the reference shape will be required. Yet, its ease of extracting additional 3D information is compatible with longitudinal monitoring required for long-term morphological improvements.

Since our results showed that there was a pattern observed during the 3 months to 12 months period, that the head resembled a sphere at 6 months and deviated after 12 months, this trend may be used to describe the normal development. Furthermore, GI may be used to identify age-specific relationships between background characteristics and head measurements that may not be visible by solely applying CI.

## Limitations

There are several limitations that need to be addressed. Firstly, there is a possibility that the direct measurements were affected by inter- and intra-rater variations as they were led by one of the four members from the laboratory and only conducted once for each infant using a tape measure. However, because this was a retrospective study, it was not possible to make modifications.

Secondly, 1,980 records included repeated visits made across different time point. However, since most of these multiple visits only involved HC measurements, they were only included in the HC comparison with the government data and not for the analyses of age and birth year impacts. Among 909 records with the complete set of measurements (HC, LT-RT, G-O), 10 represented the second visit and 1 the third visit. As the number was small, we considered these as unique individuals in this study.

In addition, this study is limited by the substantial differences observed in the distribution of infants across each monthly and birth year category, which made it challenging to satisfy the assumptions of homoscedasticity and normality generally required for linear models applied in this study.

Furthermore, given that this study did not address the accuracy of the estimations related to cranial parameters such as volume, it is important to conduct a subsequent investigation prioritizing this issue.

Lastly, this data may not represent the nationwide Japanese infant population as only those who could visit the laboratory in Tokyo in person were included. However, this may also be advantageous as geographic variations in head measurements are considered to exist within Japan in adults [20].

## Conclusions

We conducted a cross-sectional study to identify the natural course of morphological changes in the head that occur in the first year of life in healthy Japanese infants. We used a database containing head circumference (HC), glabella-occipital protuberance (G-O), and left tragion-right tragion (LT-RT) measurements obtained using tape measures and estimated height, head length and width to further estimate CI and volume. We also developed a new evaluation measure GI, which determines 3D globularity of the cranium to supplement 3D information and has the potential to be used as an additional measure for objective quantification of the infant head shape. This method requires already existing or easily collected tape measurement data and allows non-invasive long-term monitoring of healthy infant head shape in 3D. During the first year, both CI and height/half-length ratio reached their highest levels after 6 months, but CI subsequently dropped by 12 months, while height/half-length ratio remained the same. This indicated that the head displayed a front-to-back flattening with increasing vertical height at 6 months, which reverted to an elongated form after 12 months. GI showed a trend consistent with those observed in CI and height/half-length ratio, with increased resemblance to a sphere at 6 months which reversed after 12 months. Moreover, we observed differences in relations to sex. Males were likely to present higher values during the first year of life except for GI. Since the study involved a database large enough to divide infants into birth year subpopulations, we further examined the impact of birth year on head measurements and observed that secular changes were taking place over the period from 2010 to 2019, with recent birth years presenting more elongated shape. To our knowledge, few studies focused on changes that occur to head shape during the earlier months of life in healthy infants and no previous study reported the secular changes happening in infants in the early developmental stage.

## Supporting information

**S1 File. Supplementary tables.**
(PDF)

**S2 File. Supplementary figures.**
(PDF)

**S3 File. Supplementary scripts.**
(PDF)

## Acknowledgments

We would like to express our thanks to Kayo Asakawa and Yoshiko Koda for data collection; and Keiko Hirano, Tomoko Yoneyama, and Nobue Kanaya for administrative assistance.

## Author contributions

**Conceptualization:** Eujin Lee, Hama Watanabe, Fumitaka Homae, Gentaro Taga.

**Data curation:** Hama Watanabe.

**Formal analysis:** Eujin Lee.

**Funding acquisition:** Gentaro Taga.

**Investigation:** Hama Watanabe, Ryoya Saji, Fumitaka Homae, Gentaro Taga.

**Methodology:** Eujin Lee, Gentaro Taga.

**Project administration:** Hama Watanabe, Gentaro Taga.

**Resources:** Gentaro Taga.

**Software:** Eujin Lee.

**Supervision:** Hama Watanabe, Gentaro Taga.

**Validation:** Hama Watanabe, Gentaro Taga.

**Visualization:** Eujin Lee.

**Writing – original draft:** Eujin Lee.

**Writing – review & editing:** Hama Watanabe, Ryoya Saji, Fumitaka Homae, Gentaro Taga.

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
