## [Decision Letter · Decision Letter 0]

1 Sep 2025

PONE-D-25-29078Changes in Infant Head Shape: Developmental Trends During the First Year of Life and Secular Changes Observed in Recent YearsPLOS ONE

Dear Dr. Lee,

Thank you for submitting your manuscript to PLOS ONE. After careful consideration, we feel that it has merit but does not fully meet PLOS ONE’s publication criteria as it currently stands. Therefore, we invite you to submit a revised version of the manuscript that addresses the points raised during the review process.

We look forward to receiving your revised manuscript.

Kind regards,

Taher Babaee

Academic Editor

PLOS ONE

**Journal Requirements:**

1. When submitting your revision, we need you to address these additional requirements. Please ensure that your manuscript meets PLOS ONE's style requirements, including those for file naming. The PLOS ONE style templates can be found at https://journals.plos.org/plosone/s/file?id=wjVg/PLOSOne_formatting_sample_main_body.pdf and https://journals.plos.org/plosone/s/file?id=ba62/PLOSOne_formatting_sample_title_authors_affiliations.pdf 2. We note that the grant information you provided in the ‘Funding Information’ and ‘Financial Disclosure’ sections do not match.  When you resubmit, please ensure that you provide the correct grant numbers for the awards you received for your study in the ‘Funding Information’ section. 3. Thank you for stating in your Funding Statement: The study was partly supported by Japan Society for Promotion of Science Grants-in-Aid for Scientific Research (23H05425) to G.T.  Please provide an amended statement that declares *all* the funding or sources of support (whether external or internal to your organization) received during this study, as detailed online in our guide for authors at http://journals.plos.org/plosone/s/submit-now. Please also include the statement “There was no additional external funding received for this study.” in your updated Funding Statement. Please include your amended Funding Statement within your cover letter. We will change the online submission form on your behalf. 4. Thank you for stating the following in the Acknowledgments Section of your manuscript: We would like to express our thanks to Kayo Asakawa and Yoshiko Koda for data collection; and Keiko Hirano, Tomoko Yoneyama, and Nobue Kanaya for administrative assistance. The study was partly supported by Japan Society for Promotion of Science Grants-in-Aid for Scientific Research (23H05425) to G.T. We note that you have provided funding information that is not currently declared in your Funding Statement. However, funding information should not appear in the Acknowledgments section or other areas of your manuscript. We will only publish funding information present in the Funding Statement section of the online submission form. Please remove any funding-related text from the manuscript and let us know how you would like to update your Funding Statement. Currently, your Funding Statement reads as follows: The study was partly supported by Japan Society for Promotion of Science Grants-in-Aid for Scientific Research (23H05425) to G.T.  Please include your amended statements within your cover letter; we will change the online submission form on your behalf. 5. Please upload a new copy of Figure 2, 3, 4, 5, 6, 8, 9 and 10 as the detail is not clear. Please follow the link for more information: https://blogs.plos.org/plos/2019/06/looking-good-tips-for-creating-your-plos-figures-graphics/" https://blogs.plos.org/plos/2019/06/looking-good-tips-for-creating-your-plos-figures-graphics/ 6. We note that there is identifying data in the Supporting Information file. Due to the inclusion of these potentially identifying data, we have removed this file from your file inventory. Prior to sharing human research participant data, authors should consult with an ethics committee to ensure data are shared in accordance with participant consent and all applicable local laws. Data sharing should never compromise participant privacy. It is therefore not appropriate to publicly share personally identifiable data on human research participants. The following are examples of data that should not be shared: -Name, initials, physical address-Ages more specific than whole numbers-Internet protocol (IP) address-Specific dates (birth dates, death dates, examination dates, etc.)-Contact information such as phone number or email address-Location data-ID numbers that seem specific (long numbers, include initials, titled “Hospital ID”) rather than random (small numbers in numerical order) Data that are not directly identifying may also be inappropriate to share, as in combination they can become identifying. For example, data collected from a small group of participants, vulnerable populations, or private groups should not be shared if they involve indirect identifiers (such as sex, ethnicity, location, etc.) that may risk the identification of study participants. Additional guidance on preparing raw data for publication can be found in our Data Policy (https://journals.plos.org/plosone/s/data-availability#loc-human-research-participant-data-and-other-sensitive-data) and in the following article: http://www.bmj.com/content/340/bmj.c181.long. Please remove or anonymize all personal information (<specific identifying information in file to be removed>), ensure that the data shared are in accordance with participant consent, and re-upload a fully anonymized data set. Please note that spreadsheet columns with personal information must be removed and not hidden as all hidden columns will appear in the published file. 7. If the reviewer comments include a recommendation to cite specific previously published works, please review and evaluate these publications to determine whether they are relevant and should be cited. There is no requirement to cite these works unless the editor has indicated otherwise.

Reviewers' comments:

Reviewer's Responses to Questions

**Comments to the Author**

1. Is the manuscript technically sound, and do the data support the conclusions?

Reviewer #1: Partly

Reviewer #2: Partly

2. Has the statistical analysis been performed appropriately and rigorously? 

Reviewer #1: No

Reviewer #2: No

3. Have the authors made all data underlying the findings in their manuscript fully available?

Reviewer #1: Yes

Reviewer #2: Yes

4. Is the manuscript presented in an intelligible fashion and written in standard English?

Reviewer #1: Yes

Reviewer #2: Yes

5. Review Comments to the Author

**Reviewer #1:** This manuscript presents a large dataset of manual head measurements in Japanese infants to analyze changes in cranial shape during the first year of life. While the topic is relevant and the sample size robust, the study lacks methodological innovation and relies on simplistic geometric assumptions. The proposed Globularity Index (GI), though conceptually interesting, is not adequately supported by validation or comparison with 3D imaging methods. Figures and tables need improvement, and the discussion fails to explore the clinical or physiological implications of the findings. Overall, the manuscript would require substantial revision to be suitable for publication.

The following points should be addressed:

The data used in the study, was already published? If so, please, add link to data and specify the additions of this study to the already available information.

Table 1 have several mismatches and needs to be corrected or clarified:

When talking about age, the sentence “excluding 155 age groups other than 2, 3, 6, and 12 months” is repeated. Please, clarify this as the final number matches the total number of the dataset (1980 infants). Is there really an exclusion? Why? (Other ages could have been included) .

If only infants with certain ages are taken into account it should be stated and clarified at the beginning, and not repeated for each table.

In table 1 b the total number of infants is different for birth weight and birth year. There’s no reason to not show mean and median in age.

In table 1c total number of infants (sum of all tables) is 914, that does not match previous numbers.

I consider that exclusions need to be stated at the beginning, and then, used data needs to be detailed.

Subsections in table 1 are difficult to understand, I consider the information should be clarified and separated in different tables. Journal submission guidelines specify that subsections in tables should have a consistent number of columns or be divided.

Lines 518-532 are not interpretable, a better way to present the information must be found.

The resolution of all the figures needs to be improved. I cannot evaluate figure 5 as it is not visible. Explain the upper part of boxplots (Figures 3 and 4).

No discussion is made on the importance of GI and its advantages in comparison with CI.

Conclusions or discussion should include the implications of the results.

**Reviewer #2:**  First of all, I would like to thank the editors for the opportunity to review this manuscript. The study addresses an interesting and relevant topic; however, it requires substantial revisions before it can be considered for publication.”

1) Abstract

-Authors should specify the design (cross-sectional with repeated measures in a subset), inclusion/exclusion criteria, and the variables used to estimate CI/GI/volume (HC, LT-RT, G-O).

-Authors should flag upfront the non-independence limitation: records are treated as distinct individuals despite 95 infants with repeated visits.

-Authors shpuld add n by age group and effect sizes/95% CI for the main results to improve interpretability.

2) Introduction

-Authors should define more precisely the novelty of the Globularity Index (GI) relative to prior sphericity/compactness metrics and justify why the chosen approximation (ellipsoid surface via Knud Thomsen) is appropriate for infant crania, with references and/or validation.

3) Methods

3.1. Data and design

-Authors state that each record was treated as a single infant; however, 95 infants had measurements at multiple visits. This violates independence and likely underestimates the variance. Authors should reanalyze the data using mixed-effects models (with infant identification as a random effect) or, at a minimum, provide a sensitivity analysis that excludes repeated cases.

-Auhors should provide more detail on how HC, LT-RT, and G-O were collected, and define precisely the age bins (30-day windows), including ranges and the distribution per group in the text (not only in tables).

3.2. Geometric approach and assumptions

-The ellipsoidal model infers a1, a2, and h from HC, LT-RT, and G-O. Authors introduce a factor α = 0.8 to adjust LT-RT per the 10–20 system. This assumption is highly influential yet no sensitivity analysis (e.g., α = 0.75–0.85) nor external validation (a small 3D substudy) is presented. Please add both.

-3.3. GI definition and statistical handling

-GI is defined as the ratio between the surface of an ellipsoid (Knud Thomsen) and that of a volume-equivalent sphere—geometrically fine.

-Authors apply Fisher’s z transformation to GI “because values cluster near 0.9–1.0.” Fisher’s z is intended for correlation coefficients; for bounded indices, logit or arcsin-sqrt transforms or beta regression are more appropriate. Please re-analyze GI using an appropriate transformation/model and report whether inferences change.

3.4. Statistical strategy

-Authors describe a large battery of tests (Shapiro–Wilk, Bartlett, ANOVA/Welch, Kruskal–Wallis, Nemenyi, Tukey/Games–Howell, Wilcoxon/Fligner-Policello). This multiplies univariate contrasts without global type-I error control. I recommend:

Linear/multilevel models for CI/GI/volume with exact age (days), sex, year (and, if available, rater) as covariates; FDR (Benjamini–Hochberg) correction for families of comparisons.

-Restricting to 3 months may induce bias (exact age within that month, changing composition by year). Please expand with models that include continuous age and test year×age interactions.

3.5. Reproducibility

-Authors list R packages but provide no code/seed or fully reproducible analysis plan. Please deposit scripts and “session info.”

4) Results

4.1. Changes with age (2, 3, 6, 12 months)

Authors shoud add effect sizes and 95% CI for pairwise age comparisons, not only p-values.

4.2. Sex differences

-Authors ahould provide standardized effects (e.g., Cohen’s d) and 95% CI to aid interpretation.

4.3. Secular changes (2010–2019; 3 months)

-Authors report antero-posterior elongation in recent years (a1↓, a2↑; CI↓; h/a2↓) with stable volume. This section would be stronger with adjusted models (exact age in days, sex, rater if applicable), multiplicity control, and visualizations showing means + 95% CI per year.

5) Discussion

Authors should incorporate into Limitations the pseudoreplication, the inappropriate use of Fisher’s z for GI, and sensitivity to α. Current limitations mention inter/intra-observer variability and the lack of volume validation, but not these issues.

6) Tables and figures

Global assessment: low quality. Substantial improvements in design and content are required to allow unambiguous evaluation of the findings.

Specific actions:

- Readability and formatting: Increase resolution and line thickness; export in vector formats (PDF/SVG); standardize fonts and sizes.

- Axes and units: Label units (e.g., mm) clearly; use consistent scales across all figures.

- Statistical information: Add n per group, means/95% CIs, and effect sizes next to box plots.

- Clarity of the GI: Avoid displaying "GI (z-transformation)" in figures and legends; present the GI on its original scale of 0 to 1 to facilitate clinical interpretation, and if transformation is necessary, show the raw scale and report the transformation only secondarily.

Key recommendations:

- Independence: Reanalyze using mixed-effects or sensitivity models, excluding repeat visits (identification of the infant as a random effect).

- GI handling: Replace Fisher's z test with logit/arcsine-square root or beta regression; recheck inferences.

Assumption of -α = 0.8: Include a sensitivity analysis and, if possible, external validation.

- Multiplicity/modeling: Reduce reliance on numerous univariate tests; use (multi)variable models with FDR control.

- Secular trends: Model the year, adjusting for exact age (days), sex, and assessor (if applicable); report the 95% CI.

- Presentation: Improve the quality of tables/figures, add effect sizes/CIs, clarify units and n per group; avoid the "z-transformed GI" in graphs.

6. PLOS authors have the option to publish the peer review history of their article (what does this mean? ). If published, this will include your full peer review and any attached files.

**Do you want your identity to be public for this peer review?** For information about this choice, including consent withdrawal, please see our Privacy Policy .

Reviewer #1: No

Reviewer #2: No

---

## [Author Response · Author response to Decision Letter 1]

16 Jan 2026

We sincerely thank the editors and the reviewers for the insightful and thoughtful comments on our manuscript. We have revised our manuscript and would like to re-submit it for consideration for publication. We have included our point-by point responses to the reviewers’ comments in a separate file, with the file name "Response_to_Reviewers".

---

## [Decision Letter · Decision Letter 1]

10 Feb 2026

PONE-D-25-29078R1Changes in Infant Head Shape: Developmental Trends During the First Year of Life and Secular Changes Observed in Recent YearsPLOS One

Dear Dr. Lee,

Thank you for submitting your manuscript to PLOS ONE. After careful consideration, we feel that it has merit but does not fully meet PLOS ONE’s publication criteria as it currently stands. Therefore, we invite you to submit a revised version of the manuscript that addresses the points raised during the review process.

**Thank you for revising the text. As noted by Reviewer #2 in the previous review round, the study design should be explicitly stated. This study is a retrospective longitudinal observational study based on medical record review. Please ensure that this is clearly highlighted in both the abstract and methods sections.**

We look forward to receiving your revised manuscript.

Kind regards,

Taher Babaee

Academic Editor

PLOS One

**Journal Requirements:**

Reviewers' comments:

Reviewer's Responses to Questions

**Comments to the Author**

1. If the authors have adequately addressed your comments raised in a previous round of review and you feel that this manuscript is now acceptable for publication, you may indicate that here to bypass the “Comments to the Author” section, enter your conflict of interest statement in the “Confidential to Editor” section, and submit your "Accept" recommendation.

Reviewer #1: (No Response)

2. Is the manuscript technically sound, and do the data support the conclusions?

Reviewer #1: Yes

3. Has the statistical analysis been performed appropriately and rigorously? 

Reviewer #1: Yes

4. Have the authors made all data underlying the findings in their manuscript fully available?

Reviewer #1: No

5. Is the manuscript presented in an intelligible fashion and written in standard English?

Reviewer #1: Yes

6. Review Comments to the Author

**Reviewer #1:** Firstly, I would like to thanks the authors for taking into account most of my comments. I consider that the paper has been greatly improved. However, there are some further improvements I have to mention:

-Figures are still low resolution

- Figure captions need to be expanded to properly explain what the reader is seeing, some examples:

• Figures 3 and 4: As I mentioned in my previous comments the symbols at the top are not explained. The explanation is in the response but not in the manuscript.

• Fig 7. What is government? I assume it is official data but it is not mentioned.

-The authors mention that no other sphericity metrics are employed in other studies. Some studies exist and should be mentioned and discussion could be improved, for example:

Grieb, J., Barbero-García, I., & Lerma, J. L. (2022). Spherical harmonics to quantify cranial asymmetry in deformational plagiocephaly. Scientific Reports, 12(1), 167.

I agree that the method presented in the manuscript has the advantage of “Easiness of extracting information” but it should be discussed by comparison with other methods

7. PLOS authors have the option to publish the peer review history of their article (what does this mean?). If published, this will include your full peer review and any attached files.

**Do you want your identity to be public for this peer review?** For information about this choice, including consent withdrawal, please see our Privacy Policy .

Reviewer #1: No

---

## [Author Response · Author response to Decision Letter 2]

18 Feb 2026

We would like to thank the editor and the reviewer for taking your time to read the manuscript and providing valuable feedback. We have revised our manuscript following your guidance and we sincerely hope that the manuscript is now satisfactroy.

---

## [Decision Letter · Decision Letter 2]

24 Feb 2026

Changes in Infant Head Shape: Developmental Trends During the First Year of Life and Secular Changes Observed in Recent Years

PONE-D-25-29078R2

Dear Dr. Lee,

We’re pleased to inform you that your manuscript has been judged scientifically suitable for publication and will be formally accepted for publication once it meets all outstanding technical requirements.

Kind regards,

Taher Babaee

Academic Editor

PLOS One

Additional Editor Comments (optional):

Reviewers' comments:

Reviewer's Responses to Questions

**Comments to the Author**

1. If the authors have adequately addressed your comments raised in a previous round of review and you feel that this manuscript is now acceptable for publication, you may indicate that here to bypass the “Comments to the Author” section, enter your conflict of interest statement in the “Confidential to Editor” section, and submit your "Accept" recommendation.

Reviewer #1: All comments have been addressed

2. Is the manuscript technically sound, and do the data support the conclusions?

Reviewer #1: Yes

3. Has the statistical analysis been performed appropriately and rigorously? 

Reviewer #1: Yes

4. Have the authors made all data underlying the findings in their manuscript fully available?

Reviewer #1: No

5. Is the manuscript presented in an intelligible fashion and written in standard English?

Reviewer #1: Yes

6. Review Comments to the Author

Reviewer #1: (No Response)

7. PLOS authors have the option to publish the peer review history of their article (what does this mean? ). If published, this will include your full peer review and any attached files.

**Do you want your identity to be public for this peer review?** For information about this choice, including consent withdrawal, please see our Privacy Policy .

Reviewer #1: No

---

## [Editor Report · Acceptance letter]

PONE-D-25-29078R2

PLOS One

Dear Dr. Lee,

I'm pleased to inform you that your manuscript has been deemed suitable for publication in PLOS One. Congratulations! Your manuscript is now being handed over to our production team.

Kind regards,

on behalf of

Dr. Taher Babaee

Academic Editor

PLOS One